# Epidemiology of Breast Cancer in Mexican Women with Obesity as a Risk Factor

**DOI:** 10.3390/ijms231810742

**Published:** 2022-09-15

**Authors:** Shaila Cejudo-Arteaga, Miguel Ángel Guerrero-Ramos, Roberto Kuri-Exome, Erika Martínez-Cordero, Felipe Farias-Serratos, María Maldonado-Vega

**Affiliations:** 1Colonia Centro, Facultad de Medicina, Benemérita Universidad Autónoma de Puebla, 4 Sur #104, Puebla 72420, Mexico; 2Hospital Regional de Alta Especialidad del Bajío, Servicio de Oncología Médica, Blvd. Milenio #130, Colonia San Carlos La Roncha, León 37660, Mexico; 3Hospital Regional de Alta Especialidad del Bajío, Dirección de Planeación, Enseñanza e Investigación, Unidad de Enseñanza e Investigación, Blvd. Milenio #130, Colonia San Carlos La Roncha, León 37660, Mexico

**Keywords:** obesity, Mexican women, breast cancer, Ag CEA, Ag CA15-3, recurrence, macrophages

## Abstract

**Purpose**. Adipose tissue in overweight and obesity shows metabolic imbalance in the function of adipocytes and macrophages, this leads to altered regulation of hunger, lipid storage, and chronic inflammation possibly related to the development of breast cancer. **Methods**. The study was retrospective of 653 breast cancer patients treated at a tertiary care hospital. Histopathology, hormone receptors, grade, clinical stage, clinical biometry analysis, CEA and CA 15-3 antigens were analyzed. The analyses were performed at diagnosis and at the end of oncological treatments. **Results**. Mexican women studied and treated for breast cancer have an BMI of 29 from diagnosis and at the end of their cancer treatments. The average age was 52 ± 12 years, 54% in women older than 55 years. Cancer recurrence occurs in any molecular type; however, the common factor was overweight and obesity with 73% vs. 21% in normal weight patients. The most frequent tumor tissue in the population was positive hormone receptors of the luminal type (65%), HER2 (15%), and NT (15%). The analyses of macrophages/lymphocytes (M/L), CEA, and CA 15-3 antigens evaluated in women >55 and <55 years, with and without recurrence are elevated at the end of oncological treatments. **Conclusions**. The analysis of Mexican women with breast cancer showed a predominance of overweight and obesity at diagnosis and at the end of treatment. A relationship between obesity and cancer recurrence with a low response to treatment due to elevation in Ag CEA and CA 15-3 is suggested. The L/M ratio could be an indicator of inflammation related to adipose tissue since diagnosis.

## 1. Introduction

Cancer source from uncontrolled cell growth and proliferation [1]. Worldwide, the numbers for breast cancer cases are estimated at 2.2 million cases, with 685 thousand deaths from this cancer [2]. In first world countries where medical care is combined with timely detection and specific therapies, five-year survival exceeds 90%, while in low- and middle-income countries it is around 66% [3]. Among the risk factors for developing breast cancer, in addition to gender and age over 40 years, are such as hereditary history, sedentary lifestyle, overweight and obesity, changes in the reproductive pattern, consumption of alcohol, smoking habits as well as cellular and hormonal changes of the adipose tissue that lead to the first changes in the glandular tissue of the breast in the epithelial cells of the glandular ducts [4]. Adipose tissue, in addition to functioning as an energy store, has endocrine implications of the greatest relevance in the control of intake, satiety, energy expenditure, insulin resistance; additionally, it has a close interaction with the macrophages of the immune system [5,6]. The adipose tissue of postmenopausal women functions as an estrogen-producing gland [7], so that the activity of adipocytes and macrophages of adipose tissue is increased viscerally regulated by the adiponectin-leptin-estrogen axis [8]. With the overproduction of leptin and a functional increase in estrogen derived from adipose tissue, there is an increase in signals to the nucleus with the activation of oncogenic pathways such as NF-κB, AP-1, PI3K, STAT3, AKT, and JAK-1, thus leading to the release of proinflammatory proteins such as TNFα, IL-1, and IL-6, these changes are recognized as processes of low-grade local inflammation until they progress to systemic inflammation with the generation of reactive oxygen species and the promotion of tumor cell growth [9].

Hormonal deregulation of adipocytes causes the entry and accumulation of lipids with low energy expenditure and an increase in the proliferation of adipocytes, recruitment and differentiation of fibroblasts to adipocytes, with continued attraction of monocytes and macrophages from the circulation via monocyte chemoattractant protein (MCP-1) [10].

Chronic inflammation that occurs in adipose tissue is maintained by secretion of leptin, resistin, adiponectin, and visfatin from adipocytes, secretion of MCP-1, and increased release of non-esterified fatty acids; added to the fact that macrophages linked to adipose tissue secrete resistin, IL-1β, and IL-6. Together adipocytes and macrophages bound to adipose tissue secrete proinflammatory interleukins [11] that give rise to the continuity of chronic inflammation associated with obesity and hyperglycemia. In this context, the systemic response of chronic inflammation is caused by adipose tissue dysfunction, the secretion of cytokines and chemokines.

Hormones secreted from breast adipose tissue such as leptin, adiponectin, and resistin [7,8] are increased in overweight and obese people. It has been reported that leptin participates in the switching from anti-inflammatory macrophages with the M2 phenotype to the proinflammatory M1 phenotype [12,13]. It is suggested that the increase in circulating leptin could enhance the risk of developing breast cancer when the secretion of proinflammatory cytokines (IL-6, IL-1, IL-17, TNF-α and TGF-β-) is stimulated as well secreted by adipocytes [14,15,16]. The association of estrogen synthesis in adipocytes and leptin increases the risk of developing breast cancer, due to the continuous activation of aromatase. By itself, leptin appears to be enough to cause breast cancer, which would explain the premenopausal cases. During postmenopausal, the main estrogen producers are adipocytes, proportionally related between estrogen production and overactivation of aromatase in breast adipose tissue. Estrogen-mediated carcinogenesis has been identified by two main pathways binding to a receptor that activates tumor-promoting genes cAMP, EGFR and metabolites. (1) The first pathway is a mitogen-activated protein kinase (MAPK) and phosphatidyl-inositol 3 kinase (PIP3K); (2) through reactive catechol estrogenic quinones that create DNA adducts, this mechanism is estrogen receptor-independent in breast tumor tissue [8].

Once breast cancer is diagnosed, it is differentiated into 21 histological types and 4 molecular types Luminals A and B, human epidermal growth factor receptor 2 (HER2) varying in their positivity [9,17,18]. In patients with triple negative breast cancer (ER-, PR- and HER2-) they are more frequent in premenopausal women and the cases have a poor prognosis in survival due to a high recurrence and lack of response to treatment [4]. Additionally, patients with obesity and a BMI greater than 30 have a 1.35-fold higher risk compared to those with a lower BMI and have a 1.39-fold higher risk of triple-negative breast cancer.

This work describes a population of patients diagnosed with breast cancer based on their premenopausal stage (<55 years) and postmenopausal (>55 years) treated in a tertiary hospital in the period 2009–2019, and compares the molecular types of breast tumor tissue, recurrence, lymphocyte/macrophage (L/M) ratio, and oncological markers CEA and Ca 15-3 at diagnosis and after clinical treatment.

## 2. Results

### 2.1. Women under and over 55 Years of Age

Women with breast cancer were analyzed based on two groups: younger than 55 years and older than 55 years at diagnosis and at the end of cancer treatment (Table 1). The mean age of all women in the study was 52 ± 12 years with a range between 22 and 90 years.

Table 1 presents the characteristics of the two groups, showing that the BMI identifies the overweight population from diagnosis, a condition that is maintained until the end of clinical treatment. In all groups there are cases with morbid obesity as observed in the upper limits. The L/M index showed a tendency to increase at the end of the clinical treatment observed in both groups, a situation that coincides with the elevation of the CEA and CA 15-3 antigens, as expected, the average value of the antigens in all cases was found above the reference values, later, this information considered molecular type and recurrence for comparison.

Table A1 Appendix A. The frequency of mutated genes for BRCA 1/2, in the entire population studied was 5.3%, finding in the 18–55 years old group 4% of BRCA mutation 1/2. The 37.5% of cases have triple negative recipients, 33.3% LA, 17% LB, and 12.5% HER2. Based on this information, the genetic variable is not a factor in the development of breast cancer in the women studied.

### 2.2. Types of Breast Cancer in Mexican Women

The percentage distribution of breast cancer in Mexican women according to molecular type allows us to observe that the largest number of cases is found in the luminal subtypes followed by HER2 and triple negative (Figure 1A), resulting in a mostly hormone-sensitive population (Figure 1A) and in postmenopausal stage of sexual maturity (54%) (Figure 1B).

### 2.3. Percentage Stage Clinical and Tumor Grade Breast Cancer in Mexican Women

The percentage distribution of tumor grade places 53% of breast cancer in grade 2, this means a moderate differentiation in its growth; however, 31% of cases have poor differentiation with greater tumor growth (Figure 2A). This information is confirmed with the clinical stages (Figure 2B) observing very few cases of diagnosis in early phases, while the majority 33% and 43% are in stages II and III respectively, placing them in size and greater tumor extension of its origin—a condition that affects the response to treatment and recurrences; likewise, it is noteworthy that 11% in stage IV were found as cases in terminal stages and with a poor prognosis for life.

### 2.4. Characteristics Diagnostic Start and End of Treatment without Recurrence and with Recurrence of Cancer in Mexican Women

The body mass index (BMI) in women before and after clinical treatment averaged 29 ± 5.2 (m^2^/kg) and 28 ± 5.2 at the end of treatment, which represents only a reduction of one unit of BMI (Table 2). In the analysis of the population, it was found that 38% and 36% correspond to morbid obesity at diagnosis and at the end of chemotherapy treatment, respectively. Likewise, the studied population represented 73% with obesity and recurrence contrasted with 21% in the population with normal weight.

The comparison of the data of the L/M index, CEA, and CA 15-3, which was carried out in the non-recurrence and recurrence groups (Table 2) in this last condition, both at diagnosis and at the end of treatment, were found to be high. A subtle increase in the L/M index at the end of cancer treatments was observed, being noticeable in the recurrence group. The analysis in the entire population also coincides with these increases at the end of oncological treatments and are statistically different for the CEA and CA 15-3 antigens. This information is important in the population context and may indicate a part of the patients who could not respond to cancer control therapies.

### 2.5. Percentage under and over 55 Years of Age without Recurrence and with Recurrence of Cancer in Mexican Women

Table 3 shows the distribution of the molecular subtypes, the age of the women with breast cancer, and compares the women who presented recurrence. The highest recurrence was observed in the luminal A, luminal B, and HER2 molecular subtypes for the age group under 55 years, compared to older women where NT presented the highest recurrence with 45% followed by luminal A (31%). This indicates that age represents a relevant factor for recurrence in breast cancer.

### 2.6. Type of Breast Cancer in Mexican Women without Recurrence and with Recurrence

Table 4 shows the distribution of the clinical parameters of the L/M index, CEA, and CA 15-3 according to molecular type and recurrence in Mexican women. In the LA and LB molecular subtypes that presented recurrence, an increase in L/M, CEA, and CA 15-3 levels was observed at diagnosis and increased at the end of treatment, while in the HER2 and NT subtypes this increase was not observed before diagnosis.

The behavior of the L/M index and the CEA and CA 15-3 antigens evaluated in the different molecular types of tumor tissue turn out to be higher in all cases with recurrence, both at diagnosis and at the end of treatment when compared with their groups of non-recurrence. The comparison between the pairs at diagnosis and at the end was carried out if a decrease in the evaluation of CEA and CA 15-3 antigens was observed. The L/M ratio compared at diagnosis and at the end of treatment in luminal types A and B is higher by at least 1 unit at the end of treatment. It can be assumed that clinical treatment should limit tumor growth and consequently have a decrease in the release of CEA and CA 15-3 antigens; however, having a high concentration of these indicators in cases of recurrence is consistent with tumor growth.

### 2.7. Kaplan–Meier Survival Curve for Mexican Women with Breast Cancer

Figure 3 shows the Kaplan–Meier survival curve for Mexican women with breast cancer, where luminal molecular types A and B achieve a better response compared to HER2 and NT. In the women studied with breast cancer, when their response to treatment is assessed, continuous tumor growth factors are observed, as previously shown with the tumor antigens CEA and CA 15-3. In all molecular types, there were cases of recurrence due to the characteristics of tumor growth in which they are diagnosed. The data shown indicate the prevalence of overweight and obesity, which leads us to suggest this factor as a limiting factor for the control of the disease, since it is a condition of imbalance that maintains a chronic proinflammatory state that could favor continuous tumor growth, reducing the efficacy of clinical treatment.

## 3. Discussion

All the women with breast cancer analyzed retrospectively here show a high incidence of overweight and obesity since diagnosis, this condition represents a metabolic factor that is an imbalance in the storage and expenditure of lipids, insulin resistance [19], cardiovascular changes [20], increases in circulating lipids, hormonal imbalance, and increased proinflammatory cytokines [21,22]. This condition could be involved in the response to treatment. Mexican women evaluated with breast cancer in their characteristics in diagnosis, follow-up and clinical treatment influence the response and survival of patients.

Overweight and obesity as a characteristic prevails in the Mexican population and in these study women, it is not an exception. They presented an average BMI of 29 ± 5.2. This could guide how in the diagnosis most of the cases were found to be advanced grade 2 and 3, which corresponded to clinical stages II and III, with diagnoses less favorable to clinical treatment [23,24], compared to Asian populations (Taiwan) which present a BMI > 23 considered overweight, related to inflammation WAT with a BMI > 27 [24], which is observed as an epidemiological factor that can influence disease control.

Various studies [25,26] have shown that adipose tissue functions as endocrine tissue as age and maturity progress, so that, in this sense, enzymes such as aromatase represent one of the treatment pathways and response to the control of breast cancer [14,15,16], and under this condition the maturity of the woman is a factor related to the control of cancer. Taking this into consideration, it is observed that in advanced stages of breast cancer (III and IV) in patients >55 years of age and with obesity, they present high recurrence, observed in 73% in these cases compared to 21% in normal weight. The hormonal effect has a clear role and influence on the metabolic pathways at the adipose tissue level known up to now in tumor growth, such as the presence of high levels of leptin, giving a possible sequence that explains the oncogenic pathways dependent on the increase in circulating leptin that stimulate the secretion of proinflammatory cytokines, along with the overactivation of adipocytes [27].

The BRCA 1 gene mutation analysis in Mexican women population in this study was found to be 5.3%, in the 18–55 years old group 4% of BRCA mutation 1/2. The 37.5% of cases are triple negative recipients, 33.3% LA, 17% LB, and 12.5% HER2. Based on this information, the genetic variable is not a factor in the development of breast cancer in the women studied.

The comparison between factors such as age, molecular type, and recurrence analyzed with respect to the L/M index, CEA, and CA 15-3 antigens shows that the values remain elevated in luminal types in patients who presented recurrence, in the same way as in the group of women >55 years old, possibly due to the effect on estrogen production due to the imbalance of the progesterone-leptin-estrogen axis, with overweight and obesity as a common factor in the population studied. Although this condition is seen in the luminal types and there is a high survival rate in patients, this group seems to have the advantage of having a greater number of alternatives in treatment, being these hormone sensitive compared to the HER2 and NT subtypes with fewer alternatives in treatment, where there is a higher number of recurrences, especially in the group of women >55 years and NT, reflecting a poor prognosis in survival [28,29,30].

Another explanation when an increase in CEA and CA 15-3 antigens was observed after treatment in patients with recurrence suggests that this condition could be due to the growth and replication of tumor cells, or even due to toxicity in other organs during treatment. Patients with breast cancer who qualified as grade 2 and 3, in addition to the increase in tumor size, present with metastasis to adjacent and distant lymph nodes, an activity that could influence the response of the antigens in the groups with greater tumor activity. In advanced cases with clinical stage III (43%) and IV (11%) there is a low chance of treatment efficacy with limited survival in the short term.

In this work, it was observed that the molecular subtypes, regardless of the degree or clinical stage, the relationship with the L/M index, the CEA and CA 15-3 antigens in women with breast cancer presented recurrence and maintained high levels of antigens from the start, although the luminal subtypes showed greater survival due to the greater number of therapeutic alternatives, but not so for the cases of HER2 and NT.

## 4. Materials and Methods

### 4.1. Study Type

It was a retrospective, observational, descriptive, and longitudinal study registered with the ethics committee of the Hospital Regional de Alta Especialidad del Bajío CEI-064-2021.

### 4.2. Population

The study included 653 cases of women with breast cancer diagnosed and treated in the period 2009 to 2019. The economic condition of the study population corresponded to women living in rural areas with a low to medium economic level [31]. The information was obtained from the electronic registry and hospitalization records from the institutional archive. The follow-up of the cases was carried out at 5 and 10 years with the registry of recurrences.

### 4.3. Data

The BMI at diagnosis and after chemotherapy treatment took the classification of low weight less than 20, normal weight between 20 and 24.9; overweight greater than 25 and obesity greater than 30 [2].

From the clinical studies, the values of leukocytes, macrophages, CEA and CA 15-3 antigens were obtained at diagnosis and at the end of oncological treatments.

### 4.4. Statistical Analysis

The information was organized in a database to obtain means, minimums and maximums, standard deviation of the quantitative variables, and frequencies in the qualitative variables. The quantitative variables were analyzed by multiple comparison (ANOVA) Tukey test, the significant difference in the comparisons was *p* < 0.05. Chi-square tests were applied to qualitative variables to assess significance. Linear correlation tests between variables were applied.

## 5. Conclusions

The retrospective analysis of a population of Mexican women with breast cancer has a predominance of overweight and obesity both at diagnosis and at the end of treatment. They showed high recurrence (73%) with significant elevation in CEA and Ca 15-3 antigens, and a high L/M ratio from diagnosis in patients with recurrence compared to 21% recurrence in normal weight women. The highest recurrence is confirmed in triple negative patients with 45% in women older than 55 years, observed as a smaller number of therapeutic alternatives. The population studied with breast cancer differs in contrast with high overweight (BMI > 25) and obesity (BMI > 30) compared to Asian populations where their overweight is referred to as a BMI > 23 related to adipose tissue dysfunction.

## Figures and Tables

**Figure 1 ijms-23-10742-f001:**
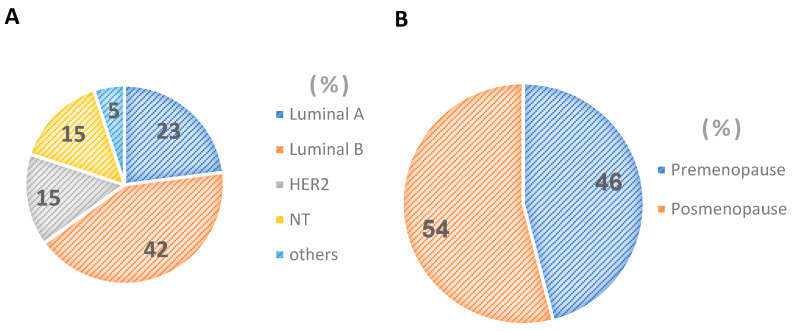
Percentage distribution of molecular types of breast cancer in Mexican women (**A**). Percentage stage menopause breast cancer in Mexican women (**B**).

**Figure 2 ijms-23-10742-f002:**
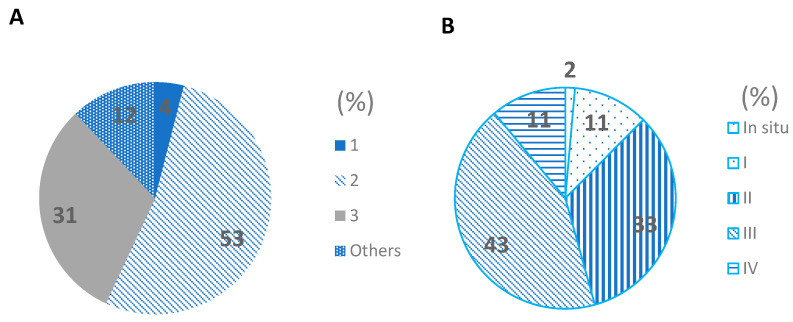
Percentage distribution of tumor grade of breast cancer in Mexican women (**A**). Percentage distribution of stage clinical of breast cancer in Mexican women (**B**).

**Figure 3 ijms-23-10742-f003:**
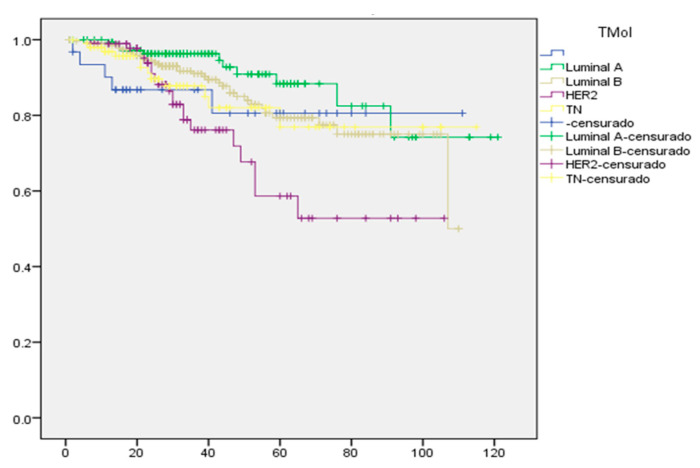
Kaplan–Meier plot of survival in Mexican women diagnosed with breast cancer. Follow-up of molecular type. Survival cumulative vs. time (month).

**Table 1 ijms-23-10742-t001:** Characteristics of Mexican women diagnosed with breast cancer and chemotherapy with respect to age group.

Characteristics	Women < 55 Years		Women > 55 Years		All Population	
	Diagnostic Start	End of Treatment	Diagnostic Start	End of Treatment	Diagnostic Start	End of Treatment
Age (years)	46 ± 8 [22–59]		68 ± 7 [59–90] *		52 ± 12 [22–90] *	
Weight (Kg)	69 ± 13 [39–133]	69 ± 14 [33–115]	67 ± 13 [34–112]	65 ± 14 [36–108]	69 ± 13 [35–133]	68 ± 14 * [33–116]
BMI (kg/m^2^)	29 ± 5.1 [17–48]	28 ± 5.2 [15–49]	29 ± 5.3 [18–45]	28 ± 5.6 [17–45]	29 ± 5.2 [15–48]	28 ± 5.2 [15–49]
Index L/M	2.74 ± 1.13 [0.01–11]	3.02 ± 1.25 [0.0–15]	2.71 ± 1.56 [0.01–10]	3.38 ± 3.19 [0–33]	2.73 ± 1.2 [0.01–11]	3.1 ± 2.1 [0–33]
CEA (ng/mL)	6 ± 34 [0–544]	26 ± 162 * [1–43]	3 ± 6 [0–2306]	5 ± 10 [1–74]	5 ± 29 [0–544]	21 ± 141 [0–2306]
CA 15-3 (U/mL)	41 ± 278 [2–4726]	65 ± 244 * [1–2773]	35 ± 56 [5–330]	32 ± 51 [4–292]	39 ± 243 [2–4723]	57 ± 214 * [1–2773]

BMI = body mass index; data are presented as the mean ± SD, the minimum and maximum values are added in square brackets. *p* < 0.05 (* statistically significant). Leukocyte/macrophage index (L/M) CEA reference value 2.5 ng/mL. Ca 15-3 reference value < 35 U/mL. Tukey multivariate ANOVA test.

**Table 2 ijms-23-10742-t002:** Characteristics of Mexican women with breast cancer diagnostic and chemotherapy by treatment and end treatment.

Characteristics	Non-Recurrence		Recurrence		All Population	
	Diagnostic Start	End of Treatment	Diagnostic Start	End of Treatment	Diagnostic Start	End of Treatment
Age (years)	52 ± 12 [22–90]		52 ± 12 [23–81] *		52 ± 12 [22–90]	
BMI (kg/m^2^)	29 ± 4.5 [17–46]	29 ± 5.1 * [16–48]	29 ± 5.6 [17–48]	28 ± 5.6 * [15–45]	29 ± 5.2 [17–48]	28 ± 5.2 * [15–49]
Index L/M	2.6 ±0.9 [0.01–8.3]	3.0 ± 1.6 * [0.0–15]	2.9 ± 1.6 [0.01–11]	3.3 ± 2.8 * [0–33]	2.7 ± 1.2 [0.009–11]	3.1 ± 2.1 * [0–33]
CEA (ng/mL)	5 ± 35 [0–544]	10 ± 58 * [0–627]	6 ± 12 [0–87]	41 ± 226 * [0–2306]	5 ± 30 [0–544]	21 ± 141 * [0–2306]
CA 15-3 (U/mL)	19 ± 22 [4–251]	65 ± 244 * [1–2773]	79 ± 416 [2–4723]	32 ± 51 * [4–292]	39 ± 243 * [2–4723]	57 ± 214 * [1–2773]

BMI = body mass index; data are presented as the mean ± SD, the minimum and maximum values are added in square brackets. *p* < 0.05 (* statistically significant). Leukocyte/macrophage index (L/M), CEA reference value 2.5 ng/mL. Ca 15-3 reference value < 35 U/mL. Tukey multivariate ANOVA test.

**Table 3 ijms-23-10742-t003:** Effect of age and recurrence observed in the different stages of breast cancer in Mexican women.

	Women < 55 Years		Women > 55 Years	
Recurrence	No (%)	Yes (%)	No (%)	Yes (%)
**LA**	75	25	68	31
**LB**	70	30	76	23
**HER2**	66	34	74	26
**NT**	84	16	55	45

**Table 4 ijms-23-10742-t004:** Distribution of the L/M response and CEA and CA 15-3 antigens regarding the molecular type of breast cancer in Mexican women.

	LA		LB		HER2		TN	
Diagnostic Start	NR	RE	NR	RE	NR	RE	NR	RE
**L/M**	3 ± 1.1 [0.01–8.3]	3 ± 1.6 [0.01–10]	3 ± 1 [0.01–6]	3 ± 2 * [0.01–11]	3 ± 1.0 [0.01–5]	3 ± 1.3 [0.01–7]	2 ± 1 [0.01–5]	3 ± 1.7 * [1.35–8]
**CEA**	2 ± 1.6 [0.2–12]	4 ± 6 * [0.2–30]	12 ± 62 [0.2–544]	9 ± 16 * [0.3–87]	4 ± 7 [0.2–41]	5 ± 7 * [0.2–33]	4 ± 10 [0.2–80]	3 ± 7 [0.4–40]
**CA 15-3**	19 ± 15 [4.7–80]	32 ± 50 * [3–2263]	18 ± 22 [5–183]	210±77 * [2–4723]	15 ± 8 [6.2–32]	27 ± 28 * [6.5–107]	21 ± 32 [4–250]	48 ± 67 * [5–330]
**End of treatment**								
**L/M**	2 ± 1.4 [0.04–9]	4 ± 4.5 * [0.02–34]	3 ± 1.8 [0.01–14]	3 ± 1.5 [2–7]	3 ± 1.6 [0.03–10]	3 ± 1.1 [2–5.5]	3 ± 2.4 [0.01–15]	3 ± 1.5 [0.16–7]
**CEA**	2 ±3.4 [0.3–23]	68 ±334 * [0.7–2306]	27 ± 105 [0.5–627]	58 ± 216 * [0.7–267]	2 ± 2.4 [0.5–15]	16 ± 51 * [0.5–293]	8 ± 25.2 [0.20–51]	4 ± 7.8 * [0.50–43]
**CA 15-3**	22 ± 36 [2–255]	122 ± 360 * [6–2223]	35 ± 87 [3.3–650]	243 ± 52 * [6–2772]	15 ± 15 [1–106]	32 ± 42 * [4–185.4]	22 ± 30 [6.3–187]	59 ± 121 * [0.5–621]

Data are presented as the mean ± SD, the minimum and maximum values are added in square brackets. *p* < 0.05 (* statistically significant). Leukocyte/macrophage index (L/M), CEA reference value 2.5 ng/mL. Ca 15-3 reference value < 35 U/mL. Tukey multivariate ANOVA test. NR = non recurrence, RE = recurrence.

## Data Availability

Third-party data restrictions apply to the availability of these data. Data were obtained from third parties and are kept confidential due to ethical policies and safeguarding the identity of study participants and are only available [from the authors/at URL] with the permission of [the treating physicians].

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
