# Peer review of "Epidemiology of Breast Cancer in Mexican Women with Obesity as a Risk Factor"

_ijms, 2022, doi:10.3390/ijms231810742_

Round 1

Reviewer 1 Report

The study by Cejudo-Arteaga et al pinpointed that adipose tissue in overweight and obesity leads to a metabolic imbalance in the function of adipocytes and macrophages, this directs to altered regulation of hunger, lipid storage, and chronic inflammation possibly conceivably linked to the growth of breast cancer. In this way, these authors analyzed the data and found that Mexican women studied and treated for breast cancer have an IBM of 29 from diagnosis and at the end of their cancer treatments. The average age was 52 ± 12 years, 54% in women older than 55 years. The common factor was overweight and obesity with 73% vs. 21% in normal weight patients. The most frequent tumor tissue in the population was positive hormone receptors of the luminal type (65%), HER2 (15%), and NT (15%). In the end, the authors concluded that the population of Mexican women with breast cancer has a predominance of overweight and obesity both at diagnosis and at the end of treatment, they showed high recurrence (73%) with significant elevation in CEA and Ca 15-3 antigens and a high L/M ratio from diagnosis in patients with recurrence compared to 21% recurrence in normal weight women. The highest recurrence is confirmed in triple-negative patients with 45% in women older than 55 years, observed as a smaller number of therapeutic alternatives.

 Comments:

The article is very interesting and relevant in the field.

The topic is original and addresses a specific gap in the field. I believe this analysis would be very useful for the clinical perspective of obesity and breast cancer.

I found the conclusion to be in line with the evidence and arguments presented.

The tables are fine.

The introduction section is a bit long. It should be reduced.

Figures 1 and 2 are fine but figure 3. It's hard to see/understand it. The authors should provide figure 3 in good quality.

Author Response

See attached file.

We appreciate your pertinent comments.

Reviewer 2 Report

Comments:

1. Any obesity marker data associated with this study?

2. On Table 1, "Women <55 years". It will be appropriate to have known age range such as 12-55.

3. On Table 1, Any data associated with BRCA1/2, ER, PR, HER2 data?

4. Please explain why choose CEA and CA 15-3 on current study? They are markers for other cancers too.

Author Response

See attached file.

We appreciate your pertinente comments.

Round 2

Reviewer 2 Report

No more comments